# Impact of self-perceived discomfort in critically ill patients on the occurrence of psychiatric symptoms in post-intensive care syndrome (PICS): A prospective observational study

Romain Ronflé[1]*, Julie Hermitant[1], Christine Conti-Zolin[1], Laurent Lefebvre[1], Thibault Helbert[1], Aurélien Culver[1], Florence Molenat[1], Baptiste Borwel[2], Mohamed Boucekine[3], Pierre Kalfon[4], Marc Leone[5]*

1 Réanimation et Surveillance Continue Médico-Chirurgicales Polyvalentes, Centre Hospitalier du Pays d'Aix, Aix-en-Provence, France, 2 Service de Psychiatrie, APHM, Hôpital Universitaire Sainte Marguerite, Marseille, France, 3 Aix-Marseille Université, APHM, EA 3279 CEReSS, School of Medicine – La Timone Medical Campus, Health Service Research and Quality of Life Center, Marseille, France, 4 Service de Réanimation, Hôpital Privé la Casamance, Marseille, France, 5 Aix-Marseille Université, APHM, Hôpital Nord, Service d'Anesthésie et de Réanimation, Marseille, France

* rronfle@ch-aix.fr; marc.leone@ap-hm.fr

## Abstract

### Background

Mental health impairments after intensive care unit (ICU) discharge include anxiety, depression, and post-traumatic stress disorder [PTSD], forming part of the post-intensive care syndrome (PICS). We assessed the effects of discomfort on the occurrence of psychiatric symptoms as a part of PICS.

### Methods

This prospective observational study conducted from September 2022 to June 2023 included all patients aged ≥ 18 years who survived an ICU stay of ≥3 days. To assess patient discomfort during the ICU stay, we used the Inconforts des Patients de REAnimation (IPREA) questionnaire. The primary outcome was the occurrence of anxiety, depression, or PTSD after ICU discharge. Secondary outcomes were the quality of life in ICU survivors and the clinical impression of physicians and psychologists to predict post-ICU psychiatric symptoms.

### Results

Of the 173 patients included initially, 109 were finally analysed. An IPREA score ≥ 13 was strongly associated with an increased risk of post-ICU psychiatric symptoms (odds ratio: 3.8, 95% confidence interval: 1.4–10.3, $p = 0.008$). The patients with post-ICU psychiatric symptoms had a reduced quality of life. The clinical impression of

**Data availability statement:** All relevant data are within the paper and its Supporting Information files.

**Funding:** The author(s) received no specific funding for this work.

**Competing interests:** the authors have declared that no competing interests exist.

physicians and psychologists at ICU discharge for the risk of psychiatric symptoms 3 months after the ICU stay was not selective.

## Conclusions

Self-perceived discomfort in ICU survivors was the most predictive factor of the development of post-ICU psychiatric symptoms.

## Introduction

Critically ill patients in intensive care unit (ICU) are exposed to stressful conditions and experience discomfort [1]. ICU survivors present an increased risk of longer-term psychopathological issues [2], primarily anxiety, depression, and post-traumatic stress disorder (PTSD). Symptoms of anxiety and depression occur respectively in 25% to 46%, and approximately 29% of patients, after discharge [3]. PTSD has largely been reported after ICU discharge [4,5]. Moreover, when symptoms of any of these psychiatric disorders are present, they occur with symptoms of the other two disorders in 65% of cases [2].

Physical, cognitive, and mental health impairment occurring after ICU discharge is known as post-intensive care syndrome (PICS) [6]. Up to a third of ICU survivors experience psychiatric impairment [7]. Significant risk factors for PICS include older age, female sex, previous mental health problems, disease severity, negative ICU experience, and delirium [8]. Unpleasant memories of real events during ICU stay may provide some protection from PICS symptoms. Conversely, when memories of delusions are prominent, PICS symptoms may be higher [1].

ICU patients are subjected to multiple sources of discomfort, including environment or types of care [9,10], all of which affect their outcome after ICU discharge. To assess the discomfort of ICU patients, the Inconforts des Patients de REAnimation (IPREA) questionnaire was established [11]. Given that only a few studies have measured the effects of self-perceived discomfort on post-ICU psychiatric symptoms, we assessed these effects on the occurrence of psychiatric symptoms compared with already known risk factors. Secondly, we analysed the effect of discomfort on the post-ICU patient's quality of life. Finally, we evaluated the clinical global impression of physicians and psychologists to predict the occurrence of post-ICU psychiatric symptoms.

## Materials and methods

### Setting

We conducted this prospective and observational study in the mixed medical–surgical ICU of Centre Hospitalier du Pays d'Aix, Aix-en-Provence, France. The study was approved by the Ethics Committee of Rouen University, France (approval number: 2022-A01063-40). We obtained the patient's or relatives' written consent for using these data at this ICU discharge. The study adheres to the Strengthening the Reporting of Observational Studies in Epidemiology (STROBE) recommendations for cohort

studies. Our study protocol is available as supporting information from the authors. The authors confirm that all ongoing and related trials for this study are registered (clinicaltrial.gov; NCT06238557).

## Patients

From 8th September 2022 to 14th October 2023, all patients aged ≥18 years who survived an ICU stay of ≥3 days were deemed eligible for inclusion in this study. We excluded patients under guardianship, patients with mental disability (neurodevelopmental, intellectual, or cognitive), patients who did not understand French sufficiently to be questioned, and patients included in another interventional study. We also excluded patients who denied the use of personal data or did not respond after 3-month ICU discharge.

## Study design

First, we collected on ICU discharge the IPREA score, memories during ICU stay, and self-perceived frightening experiences from ICU admission to ICU discharge. The clinical global impression of the physician and psychologist in charge of the patient about the risk of psychiatric components of PICS were assessed simultaneously. Then 3 months after ICU discharge, the questionnaire of anxiety, depression, and PTSD was conducted via conference call or through self-assessment. Finally, the patient's quality of life after ICU discharge was investigated.

## Measurement

The included patients' clinicodemographic data were collected from electronic medical records: demographics (sex, age, preexisting psychopathology (depressive disorder, anxiety disorder, schizophrenia, bipolar disorder, post-traumatic stress disorder), condition of admission (surgical or nonsurgical), health status before the ICU stay (the Knaus score), and health status at ICU admission (Simplified Acute Physiology Score of 2 [SAPS2] and Sepsis-related Organ Failure Assessment [SOFA] score). We also recorded any stressful procedures that the patients underwent during their ICU stay: invasive mechanical ventilation, noninvasive ventilation, tracheotomy, renal replacement therapy, central venous or arterial catheter insertion (and number), nasogastric tube insertion, bronchoscopy, digestive endoscopy, chest drainage, external electrical shock, lumbar puncture, intrahospital transport, and external defibrillation. The number of stressful procedures was recorded as follows: one point for each procedure (among the 14 aforementioned procedures). The number of days of benzodiazepine and/or neuroleptic medication was used to report on acute agitation. Finally, ICU length of stay, memories during sedation (remember information during medically induced coma), and the occurrence of frightening experiences were also recorded.

To assess patient discomfort during the ICU stay, we calculated the self-reported discomfort score by using the 18-item French-language IPREA questionnaire (S1 Table), which was validated by the IPREA group in a large sample of critically ill patients hospitalised in 34 French ICUs [12].

The IPREA score was calculated as the mean of the 18 scores reported for each item multiplied by 10, yielding a score ranging from 0 to 100, with higher scores indicating increased discomfort. Any physician of our ICU could administer the questionnaire provided and underwent specific training. The questions were asked in random order to reduce bias.

## Outcomes

The primary outcome was the psychiatric component of PICS, defined as the occurrence of anxiety, depression or PTSD. The occurrence of at least one of the three psychiatric symptoms is sufficient to complete the primary outcome. Anxiety and depression were evaluated using a validated tool: the 14-item Hospital Anxiety and Depression Scale (HADS) [13], which includes 7 questions each on depression and anxiety, each scored on a 4-point Likert scale ranging from 0 (no impairment) and 3 (severe impairment). A cutoff score of ≥8 for either anxiety or depression subscales indicated the presence of the respective condition. We applied HADS boundaries for mild, moderate, and severe symptoms to those

exhibiting symptoms. PTSD was assessed using the PTSD Checklist for DSM-5 (PCL-5), a 20-item self-report measure of the 20 DSM-5 symptoms of PTSD [14]. Included in the scale are four domains consistent with the four criteria of PTSD in DSM-5: Reexperiencing (criterion B), Avoidance (criterion C), Negative Alterations in Cognition and Mood (criterion D), and Hyperarousal (criterion E). To define a case, we treated each item rated as 2 = "Moderately" or higher as a symptom endorsed and then followed the DSM-5 diagnostic rule: ≥1 Criterion B item (questions 1–5), ≥ Criterion C item (questions 6–7), ≥2 Criterion D items (questions 8–14), and ≥2 Criterion E items (questions 15–20).

The secondary outcome was the quality of life after 3-month ICU discharge, which was evaluated using the World Health Organisation Quality of Life-BREF (WHOQOL-BREF) questionnaire. The score is calculated by adding up the responses to the questions and transforming them to a scale from 0 to 100, with higher scores indicating a better quality of life. We preferred this questionnaire over the SF-36 as it includes items on the environment. Moreover, we developed a risk score to predict post-ICU psychiatric symptoms based on the major risk factors identified in our cohort. Finally, the clinical impression of the ICU physician and psychologist at ICU discharge for the risk of psychiatric symptoms 3 months after ICU stay was studied. At the ICU discharge, ICU physicians and psychologists met the patient for a clinical interview to assess the occurrence of psychiatric symptoms or not (anxiety, depression or PTSD; associated or not) before scoring his or her self-perceived discomfort. They used clinical experience and patients' clinical data only (health status at ICU admission, condition of admission, number of stressful procedures, ICU length of stay, memories during sedation and frightening experiences).

## Statistical analysis

Continuous variables are expressed as means ± standard deviation or median (interquartile range) and were compared using Student's t test. Categorical variables are expressed as frequency and percentage and were compared using the chi-square test or Fisher's exact test. Multivariate logistic regression models were used to identify independent risk factors for psychiatric symptoms. Multivariate logistic regression model was constructed following the commonly recommended guideline of maintaining approximately 10 outcome events per predictor variable to mitigate the risk of model overfitting and excessive complexity [15]. The selection of variables was based on a univariate analysis threshold ($p < 0.2$) combined with clinical relevance and prior evidence on known confounders. Results are presented as odds ratios (ORs) with 95% confidence intervals (CIs). In all analyses, two-tailed $p < 0.05$ was considered significant. Next, we built a risk score for post-ICU psychiatric symptoms with the significant parameters. Variables were categorized to improve clinical interpretability and define homogeneous risk groups, with cutoff values determined using receiver operating characteristic (ROC) curves and the Youden index. ROC curve analyses were performed to assess the effectiveness of our score in predicting psychiatric symptoms. The cutoff for prediction of psychiatric symptoms was also determined using ROC curves and the Youden index. We conducted internal validation of our score in the cohort with values of sensitivity, specificity, negative predictive value, and positive predictive value. To evaluate the stability and robustness of our findings, we conducted sensitivity analyses by testing multiple model specifications with varying covariate selections. This approach allowed us to confirm the consistency and reliability of our main conclusions across different analytical frameworks. AIC and BIC values were also reported for each multivariate model to assess model fit and complexity. A size-proportional Venn diagram was made to represent the psychiatric issue components (PTSD, anxiety, and depression). Agreement between the patient's feeling of distress as assessed by the physician or psychologist at ICU discharge to the occurrence of psychiatric symptoms at 3-months was evaluated using the Cohen's kappa coefficient, which measures inter-rater reliability beyond chance, with p-values assessing its significance against zero. Taking into account the previous hypotheses, to highlight an odds ratio of 1.64 per unit of standard deviation of the IPREA score (i.e., an increase in risk of 4.5% per unit of IPREA score), with an alpha risk of 5% and a power of 80%, the minimum number of patients was determined to be 153; therefore, the number of patients expected to be included was 183 to account for loss in follow-up (20%). This number of patients is expected to show an area under the curve (AUC) of 0.7 with a 95% CI of 0.19 (153 patients required: 46 PICS and 107 no PICS).

 

All analyses were performed with available data using SPSS v19 (IBM) and R statistical software.

## Results

During the study period, 173 patients were included. Of these inclusions, 64 were excluded from the analysis for the following reasons: 11 died during the follow-up, 7 had significant missing data, and 46 did not respond. Finally, the data of 109 patients were analysed (S1 Fig). Table 1 presents their clinicodemographic characteristics.

We observed no statistical difference between the included patients and those lost to follow-up (sex, age, or preexisting psychiatric disorder). We reported an average of four stressful invasive procedures for each patient. The median self-perceived discomfort score derived from IPREA was 20 ± 14. The three highest scores were anxiety (3.8 ± 3.4), thirst (3.0 ± 3.5), and shortness of breath (3.0 ± 3.3, S2 Table).

**Table 1. Patients' characteristics.**

| Variables | Cohort (n = 173) |
|---|---|
| **Demographics** | |
| Age, y, median (Q25-Q75) | 67 (51-74) |
| Male sex, *n* (%) | 117 (68) |
| Knaus Score, *n* (%) | |
| ▪ Normal health status | 53 (31) |
| ▪ Moderate activity limitation | 80 (46) |
| ▪ Severe activity limitation | 39 (22) |
| ▪ Bedridden patient | 1 (1) |
| Preexisting psychopathology, *n* (%) | 23 (13) |
| **Diagnosis and severity** | |
| Reason for ICU admission, *n* (%) | |
| ▪ Surgical | 40 (23) |
| ▪ Non surgical | 133 (77) |
| SAPS2 score, median (Q25-Q75) | 38 (26-50) |
| SOFA score on admission, median (Q25-Q75) | 4 (2-7) |
| **Evolution** | |
| ICU length of stay, d, median (Q25-Q75) | 6 (4-10) |
| **Organ failure** | |
| Mechanical ventilation, *n* (%)* | 112 (65) |
| ▪ Invasive | 68 (39) |
| ▪ Non invasive | 79 (46) |
| Catecholamines, *n* (%) | 77 (45) |
| Renal remplacement therapy, *n* (%) | 26 (15) |
| Density of stressful procedures, mean ± SD | 4 ± 2 |
| **ICU experience** | |
| Score derived from IPREA questionnaire, mean ± SD | 20 ± 14 |
| Presence of a subjective trauma, *n* (%) | 55 (32) |
| Delusion, *n* (%) | 14 (8) |
| Use of benzodiazepines and/or neuroleptics for restless, *n* (%) | 81 (47) |

Values are expressed as number (%), mean ± SD and median (Q25-Q75). SOFA Sepsis-related Organ Failure Assessment, SAPS2 Simplified Acute Physiology Score of 2, ICU Intensive Care Unit, IPREA inconfort des patient de reanimation, *Some patients had both.

The primary outcome was observed in 44 (40%) patients of our cohort study. Twenty (48%) patients had at least two psychiatric symptoms (Fig 1). The median HADS anxiety, depression, and PCL-5 total scores were respectively 5 (2–8), 3 (2–7), and 9 (4–16) (S1 Fig, Flowchart).

From the univariate analysis, we selected several factors for predicting psychiatric symptoms: age, IPREA score, female sex, preexisting psychiatric disorder and health status with limitation of activity ($p < 0.2$). To predict post-ICU psychiatric symptoms, the best threshold for IPREA score was 13 (sensitivity: 82%, specificity: 46%, PPV: 51%, NPV: 79%, and Youden: 0.28), and the best threshold for age was 52 years (sensitivity: 37%, specificity: 85%, PPV: 62%, NPV: 67%, and Youden: 0.22). In multivariate analysis, an IPREA score $\geq 13$ was strongly associated with an increased risk of post-ICU psychiatric symptoms (OR: 3.8, 95% CI: 1.4–10.3, $p = 0.008$) (Table 2).

We then built a predictive score of psychiatric symptoms with three variables: IPREA score $\geq 13$, preexisting psychiatric disorder, and age $\leq 52$ years, with each variable counting as one point. A risk score $\geq 2$ was the best cutoff associated with psychiatric symptoms for ICU survivors (OR: 6.2, 95% CI: 2.4–16, $p < 0.001$) with an AUC of 72% (63–82%) (sensitivity: 47%; specificity: 88%, PPV: 71%, NPV: 71%, and Youden: 0.34) (Fig 2).

No difference was found in patient's characteristics between those achieving the primary outcome and those not achieving the primary outcome, except for the IPREA questionnaire and preexisting psychiatric disorders (Table 3).

The patients for whom the primary outcome was observed had their quality of life reduced according to the WHOQOL-BREF questionnaire: 49% for physical health, 56% for psychological health, 63% for social relationships, and 66% for environmental health ($p < 0.01$ for each, Table 3). Finally, the clinical impression of the physician and psychologist at ICU discharge for the risk of psychiatric symptoms 3 months after ICU stay was not selective ($\kappa = 0.04$ and 0, $p = 0.7$ and 0.96, respectively).

## Discussion

Our study explored predictive factors of psychiatric symptoms in PICS 3-months after an ICU stay. Our findings suggest that self-perceived discomfort during ICU stay is associated with an increased risk of psychiatric symptoms. Preexisting psychiatric disorders and age $\leq 52$ years were associated with an increased risk of anxiety, depression, or PTSD. Thus, we proposed a score featuring self-perceived patient discomfort, preexisting psychiatric disorders, and age $\leq 52$ years to predict the occurrence of psychiatric symptoms and long-term outcomes in ICU survivors. In addition, we suggested that patients who developed psychiatric symptoms after an ICU stay had a significant decrease in the quality of life.

Our findings are in line with those reported in previous studies. Kalfon et al. developed and validated the 18-item IPREA score for patients' self-perceived ICU discomfort [11,12]. In our study, the mean IPREA score was low, but this was in line

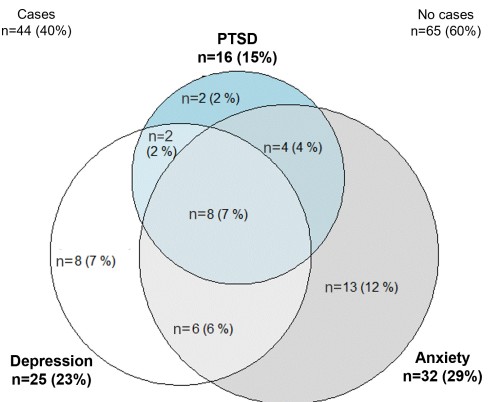

**Fig 1. Occurrence of psychiatric symptoms 3-months after ICU stay in ICU survivors.** PTSD, post-traumatic stress disorder.

**Table 2. Effect of variables on the PICS occurence at 3 months in multivariate analysis.**

| | OR (95% CI) | p value |
|---|---|---|
| **Model 1 (AIC = 28, BIC = 39)** | | |
| IPREA score ≥ 13 | 3.9 (1.5 - 10.3) | 0.007 |
| Psychiatric history | 3.5 (0.9 - 13.3) | 0.06 |
| Age ≤ 52 y | 2.6 (1 - 6.8) | 0.06 |
| **Model 2 (AIC = 42, BIC = 55)** | | |
| IPREA score ≥ 13 | 3.8 (1.4 - 10.3) | 0.008 |
| Psychiatric history | 3.6 (0.9 - 14) | 0.06 |
| Age ≤ 52 y | 3.3 (1.1 - 9.8) | 0.03 |
| Health status with limitation of activity[a] | 1.9 (0.6 - 5.4) | 0.2 |
| **Model 3 (AIC = 58, BIC = 74)** | | |
| IPREA score ≥ 13 | 3.4 (1.2 - 9.4) | 0.02 |
| Psychiatric history | 3.5 (0.9 - 13.3) | 0.07 |
| Age ≤ 52 y | 3.7 (1.2 - 11.2) | 0.02 |
| Health status with limitation of activity[a] | 2 (0.7 - 6) | 0.2 |
| Female sex | 1.7 (0.7 - 4.4) | 0.3 |

[a]Determined by the Knaus score, ranging from moderate limitation of activity to bedridden patient.

PICS, post intensive care syndrome; OR, odds ratio; CI, confidence interval.

Age ≤ 52 years was also associated with increased risk (OR: 3.3, 95% CI: 1.1–9.8, p = 0.03).

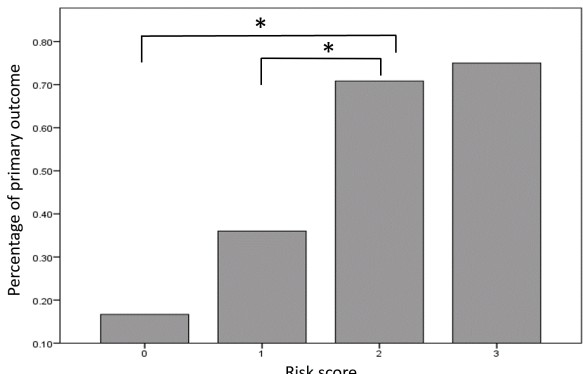

**Fig 2. Risk score of psychiatric symptoms 3 months after ICU stay in ICU survivors.** *p < 0.05.

with other studies [11,16]. Self-perceived discomfort appears to be a useful tool which can predict psychiatric symptoms and should be considered by healthcare workers to identify sources of discomfort, enabling them to adapt environment and care to minimise the risk of PICS. The implementation of specific measures that included a programme of reduction discomfort by healthcare teams resulted in a significant decrease in the IPREA score [16]. The reduction of self-perceived discomfort by a adjusted multicomponent programme could also reduce the prevalence of PTSD symptoms in ICU survivors [17]. As predicted, self-perceived discomfort measured at the end of the ICU stay was increased in ICU survivors with psychiatric symptoms at 3 months after ICU discharge. In another study, patients with PTSD symptoms 1 year after the ICU stay had higher overall discomfort scores than those without such symptoms [17].

In our study, age ≤52 years increased the occurrence of psychiatric symptoms. At first, increased age was an independent predictor of a low discomfort score in multivariate analysis [18]. Kalfon et al. observed a significantly lower age for

**Table 3. Patients' characteristics comparison between observed primary outcome group and opposite group.**

| Variables | Observed primary outcome group (n = 44) | Opposite group (n = 65) | *p* value |
|---|---|---|---|
| Age, y, mean ± SD | 59 ± 17 | 65 ± 13 | 0.12 |
| Male sex, n (%) | 28 (64) | 49 (75) | 0.20 |
| Health status with limitation of activity, n (%) | 32 (73) | 46 (71) | 0.99 |
| Preexisitng psychopathology, n (%) | 9 (21) | 4 (6) | **0.03** |
| SAPS2 score, mean ± SD | 39 ± 17 | 39 ± 14 | 0.96 |
| SOFA score, mean ± SD | 5 ± 3 | 5 ± 3 | 0.82 |
| ICU length of stay, d, mean ± SD | 11 ± 13 | 12 ± 16 | 0.18 |
| Mechanical ventilation, n (%) | 17 (39) | 27 (42) | 0.84 |
| Density of stressful procedures, mean ± SD | 3 ± 2 | 4 ± 3 | 0.54 |
| Score derived from IPREA questionnaire, mean ± SD | 23 ± 12 | 20 ± 13 | **0.05** |
| Days of benzodiazepines/ neuroleptics use, d, mean ± SD | 6 ± 8 | 6 ± 4 | 0.28 |
| Presence of a trauma, n (%) | 18 (41) | 21 (33) | 0.42 |
| Delusion, n (%) | 5 (11) | 4 (6) | 0.48 |
| Physical health, mean ± SD | 49 ± 18 | 64 ± 16 | **<0.001** |
| Psychological health, mean ± SD | 56 ± 20 | 70 ± 14 | **<0.001** |
| Social relationships, mean ± SD | 63 ± 22 | 75 ± 14 | **0.003** |
| Environmental health, mean ± SD | 66 ± 17 | 80 ± 14 | **<0.001** |

Values are expressed as number (%), mean ± SD and median (Q25–Q75).

SOFA, Sepsis-related Organ Failure Assessment; SAPS2, Simplified Acute Physiology Score of 2; ICU, Intensive Care Unit; IPREA, discomfort intensive care patients.

patients with PTSD symptoms at the 1-year follow-up [17]. A systematic review of ICU survivors reported that younger age predicted post-ICU PTSD [19]. High levels of PTSD symptoms were less likely to occur in older patients, with symptoms declining after the age of 50 years [20].

Preexisting psychiatric disorders are likely to increase the risk of psychiatric symptoms in PICS. Preexisting anxiety has shown to be a risk factor for PTSD in urban populations [21]. A meta-analysis found that only pre-ICU comorbid psychopathology was a risk factor for PTSD [4]. Patel et al. confirmed the identification of this pre-ICU risk factor [5]. Indeed, they reported both preexisting depression and preexisting PTSD as risk factors for PTSD in ICU survivors. Screening by health care teams for preexisting psychiatric status of ICU patients could be a first step.

We developed a score for different variables readily available in daily practice to assess the risk of psychiatric symptoms in ICU survivors. In the same way, Wade et al. created the intensive care psychological assessment tool to detect acute psychological distress and assess the risk of future psychological morbidity in critically ill patients [22].

Whether illness severity or invasive procedure serve as ICU risk factors for psychiatric components of PICS remains controversial. In our study, illness severity as reported through different scores (SAPS2 and SOFA) was not associated with the occurrence of psychiatric symptoms. Broomhead et al. confirmed that illness severity at ICU admission was consistently not a risk factor for PICS [23]. However, illness severity was associated with post-ICU anxiety, depression, and PTSD in other studies [3,8]. As predicted, we did not find any impact of either type or number of invasive procedures on the development of psychiatric symptoms. Fear experienced acutely during these invasive procedures and frightening memories about them seem to be the most important issue.

Our primary endpoint was the occurrence of anxiety, depression, or PTSD, and 48% of the our patients with one of these conditions had another, consistent with the finding of Hatch et al. in a UK-wide prospective cohort study [2]. Moreover, we observed a significant effect of post-ICU anxiety, depression, or PTSD on the quality of life of ICU survivors. Other studies highlighted the adverse effects of post-ICU psychiatric symptoms on the long-term outcomes of ICU

survivors. Post-ICU depressive and PTSD symptoms may be associated with both physical and mental health aspects of quality of life [19,24,25]. Through these various data, we attempted to underline the reliability of our primary outcome.

Ours is the first study to assess the clinical impressions of physicians and psychologists in charge of ICU patients for the risk of psychiatric components in PICS. Notably, we reported no correlation between clinical global impressions and the occurrence of psychiatric symptoms in ICU survivors. We strongly highlighted the benefit of using variables to help physicians and psychologists predict psychiatric outcomes in ICU survivors.

Our study has several limitations. First, we excluded more patients than expected due to an increase in lost follow-up data. We observed 46 (26%) excluded patients for missing data, due to no respond after 3-month of follow up (20% initially planned). However, the proportion of included patients with complete follow-up data at 3-months was slightly over 50%? This was comparable to many studies on post-ICU psychiatric symptoms [4,5,17]. Moreover, we observed 44 (25%) patients for whom the primary outcome was reached. That was in line with our required number of patients to show an AUC of 0.7 with a 95% CI of 0.19 (153 patients required: 46 with PICS and 107 without PICS). Secondly, female sex was not associated with an increased risk of psychiatric symptoms in multivariate analysis. Several studies have reported that women are at higher risk of PTSD symptoms than men. Preexisting psychiatric disorders were probably underestimated, explaining the lack of difference in our multivariate analysis. Thirdly, we did not use a PCL-5 cutoff to determine post-ICU PTSD symptoms. Several studies selected a PCL-5 cutoff score of 31 or 33 [26]. Half of our post-ICU patients with PTSD had a PCL-5 score of ≥31. Physical health and function, such as ICU-acquired weakness, may also affect mental health outcomes and may limit the understanding of the relationship between self-perceived discomfort and psychiatric outcomes [25]. Baseline of patient quality of life may have influenced the development of psychiatric outcomes after the ICU stay. Future research to explore the interplay between quality of life, ICU-related discomfort and psychiatric outcomes would offer a more holistic understanding of post-ICU recovery. Finally, frightening memories or delusions were not correlated with post-ICU psychiatric symptoms. We explain this by self-perceived patient reporting data instead of standardised reporting data. Future studies should consider these limitations to validate the efficacy of our predictive factors.

## Conclusion

Self-perceived discomfort in ICU survivors was the most predictive factor for developing post-ICU psychiatric symptoms in PICS. Our data indicated that a score combining age ≤52 years, preexisting psychiatric symptoms, and self-perceived discomfort could predict the occurrence of anxiety, depression, or PTSD 3 months after ICU discharge. The increasing number of ICU survivors means that future studies and a continuous quality health care improvement strategy must focus on improving long-term patients' outcomes and quality of life after ICU stay.

## Supporting information

**S1 Fig. Flowchart.**
(TIF)

**S1 Table. The French IPREA questionnaire for assessing self-reported discomforts perceived by the critically ill patients, original version.**
(DOCX)

**S2 Table. Self-reported discomforts perceived by the critically ill patients.**
(DOCX)

**S1 File. Renamed 71e78.**
(XLSX)

## Acknowledgments

The authors are indebted to the nursing staff and psychologist of polyvalent intensive care units for providing the best care to their patients.

## Author contributions

**Conceptualization:** Romain Ronflé, Julie Hermitant.

**Formal analysis:** Julie Hermitant, Mohamed Boucekine.

**Investigation:** Christine Conti-Zolin, Laurent Lefebvre, Thibault Helbert, Aurélien Culver, Florence Molenat.

**Methodology:** Romain Ronflé, Mohamed Boucekine.

**Supervision:** Marc Leone.

**Visualization:** Romain Ronflé.

**Writing – original draft:** Julie Hermitant.

**Writing – review & editing:** Baptiste Borwel, Pierre Kalfon.

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
