## [Decision Letter · Decision Letter 0]

14 Jan 2025

Dear Dr. Ronflé,

Thank you for submitting your manuscript to PLOS ONE. After careful consideration, we feel that it has merit but does not fully meet PLOS ONE’s publication criteria as it currently stands. Therefore, we invite you to submit a revised version of the manuscript that addresses the points raised during the review process.

**ACADEMIC EDITOR:**

We look forward to receiving your revised manuscript.

Kind regards,

Michihiro Tsubaki

Academic Editor

PLOS ONE

2. In this instance it seems there may be acceptable restrictions in place that prevent the public sharing of your minimal data. However, in line with our goal of ensuring long-term data availability to all interested researchers, PLOS’ Data Policy states that authors cannot be the sole named individuals responsible for ensuring data access (http://journals.plos.org/plosone/s/data-availability#loc-acceptable-data-sharing-methods).

4. We notice that your supplementary figures are uploaded with the file type 'Figure'. Please amend the file type to 'Supporting Information'. Please ensure that each Supporting Information file has a legend listed in the manuscript after the references list.

Reviewers' comments:

Reviewer's Responses to Questions

**Comments to the Author**

1. Is the manuscript technically sound, and do the data support the conclusions?

Reviewer #1: Yes

Reviewer #2: Yes

2. Has the statistical analysis been performed appropriately and rigorously?

Reviewer #1: No

Reviewer #2: N/A

3. Have the authors made all data underlying the findings in their manuscript fully available?

Reviewer #1: Yes

Reviewer #2: Yes

4. Is the manuscript presented in an intelligible fashion and written in standard English?

Reviewer #1: Yes

Reviewer #2: Yes

Reviewer #1: Thank you for the opportunity to review your study titled:

"Impact of self-perceived discomfort in critically ill patients on the occurrence of psychiatric symptoms in post-intensive care syndrome (PICS): A prospective observational study."

This study addresses an important topic with significant clinical relevance. However, several critical aspects of the methodology, statistical analysis, and results require further clarification and refinement. Specifically, issues related to the validation of measurement tools, transparency in statistical methods, and the rationale behind key analytical choices need to be addressed to ensure the robustness and reproducibility of the findings.

The comments provided below highlight areas requiring substantial revision. Addressing these concerns will significantly improve the clarity, accuracy, and scientific rigor of the manuscript.

A major revision is necessary to address these methodological and analytical concerns effectively.

Please refer to the detailed comments in the sections below.

Outcomes

Validation of French Version Scales:

It should be clarified whether the French versions of the scales used in this study have been validated. References to French domestic publications or conference presentations would be acceptable to support this validation.

Clinical Impression at ICU Discharge:

The study mentions the clinical impression of ICU physicians and psychologists at discharge to predict psychiatric symptoms three months post-ICU stay. However, the methodology lacks sufficient detail regarding how these predictions were made. Specifically:

Were predictions based solely on clinical interviews or supported by structured data?

What criteria or scoring methods were used to assess the risk?

These details are essential for evaluating the reliability and reproducibility of the findings.

Statistical Analysis

T-test vs. Welch’s Test:

Welch's t-test is generally considered more robust when there are concerns about equal variances between groups. The manuscript does not address whether the assumption of equal variances was tested. It would be more appropriate to use either Welch's t-test exclusively or clearly justify the choice of the standard t-test.

Categorical Variables:

For categorical variables, Fisher's exact test is preferable due to its robustness, especially for smaller sample sizes. Using only Fisher’s exact test could simplify and strengthen the statistical approach.

Multivariate Model Construction:

It is unclear whether the multivariate models were built using a forced-entry method or another approach (e.g., stepwise selection). The manuscript should explicitly state the method used for model construction.

Results

Cutoff Values:

The rationale for using cutoff values instead of continuous variables is not provided. Continuous data might offer more granularity and statistical power. Additionally, the precision of the cutoff values seems questionable and should be justified statistically.

Reason for Three Models:

The reason for constructing three separate models is not explained. Furthermore, no model fit indices (e.g., AIC, BIC, or R-squared) are reported, making it impossible to determine which model performs best.

Expression of Primary Outcome:

The phrase "achieved primary outcome" implies a positive connotation, which might not be appropriate in this context. A more neutral expression, such as "primary outcome was assessed" or "primary outcome was observed," would be preferable.

Statistical Method for Clinical Impression:

The statistical method used to analyze "the clinical impression of the physician and psychologist" is not mentioned in the Methods section. This lack of clarity prevents a proper evaluation of the robustness of the results.

Reviewer #2: Review Comments to the Author

General Comment

The manuscript provides valuable insights into the role of self-perceived discomfort during ICU stays as a predictive factor for post-ICU psychiatric disorders. The methodology is sound, and the statistical analyses are appropriate for the study objectives. However, some areas require further clarification and refinement to enhance the robustness and interpretability of the findings.

Specific Comments

Sample Size and Attrition

The final analysis included 109 participants, which is below the minimum required sample size of 153 calculated during the study design phase. While the study provides meaningful results, the reduced sample size may limit the statistical power and generalizability of the findings.

Furthermore, the high attrition rate (37%, with 64 participants excluded due to loss to follow-up or incomplete data) is a concern. Although the authors report no significant differences between included and excluded participants, the impact of this attrition on the representativeness of the cohort should be discussed in more detail. It is recommended that future studies adopt measures to minimize attrition, such as more robust follow-up mechanisms or alternative methods for data collection.

Adjustment for Confounders

The study effectively uses multivariable logistic regression to adjust for confounding factors, including age, pre-existing psychiatric disorders, and IPREA scores. However, quality of life (QoL) was evaluated as a secondary outcome and not included as a potential confounding factor in the analysis. Considering that QoL is closely associated with both ICU experiences and psychiatric outcomes, its exclusion as a covariate may limit the understanding of its role in mediating or moderating the relationship between self-perceived discomfort and psychiatric outcomes. Future studies could benefit from incorporating QoL as a covariate to disentangle its effects and provide more nuanced insights.

Role of QoL in the Analysis

While the study highlights the impact of psychiatric symptoms on QoL, it does not assess whether baseline QoL or changes in QoL might influence the development of psychiatric symptoms. Including QoL as a covariate or mediator in the statistical models could improve the comprehensiveness of the findings and clarify causal pathways.

Clarification in the Discussion

The discussion could be strengthened by acknowledging these limitations, particularly the small sample size, high attrition rate, and the exclusion of QoL as a covariate. Additionally, providing recommendations for future research to explore the interplay between QoL, ICU-related discomfort, and psychiatric outcomes would offer a more holistic understanding of post-ICU recovery.

Conclusion

The manuscript makes a significant contribution to understanding post-ICU psychiatric disorders, but addressing the sample size limitations, high attrition rate, and the interplay between QoL and psychiatric outcomes would improve its clinical relevance. I recommend revising the discussion to acknowledge these limitations and consider their implications for future research. These enhancements would increase the robustness of the findings and their applicability in clinical settings.

**Do you want your identity to be public for this peer review?** For information about this choice, including consent withdrawal, please see our Privacy Policy

Reviewer #1: No

Reviewer #2: No

---

## [Author Response · Author response to Decision Letter 0]

12 Mar 2025

Response to Reviewer #1

Dear Reviewer,

Your comment about our manuscript made us possible to greatly improve him.

We hope that these responses will satisfy you.

Sincerely yours

Reviewer #1:

This study addresses an important topic with significant clinical relevance. However, several critical aspects of the methodology, statistical analysis, and results require further clarification and refinement. Specifically, issues related to the validation of measurement tools, transparency in statistical methods, and the rationale behind key analytical choices need to be addressed to ensure the robustness and reproducibility of the findings. The comments provided below highlight areas requiring substantial revision. Addressing these concerns will significantly improve the clarity, accuracy, and scientific rigor of the manuscript. A major revision is necessary to address these methodological and analytical concerns effectively.

We apologize if our manuscript suffers from several critical aspects of the methodology, statistical analysis and result. We carry out a new complete proofreading and we give you a total revised and cleaned manuscript.

Reviewer #1: Outcomes

Validation of French Version Scales:

It should be clarified whether the French versions of the scales used in this study have been validated. References to French domestic publications or conference presentations would be acceptable to support this validation.

At first, The IPREA study group used the 16-item version of the IPREA questionnaire to explore self-perceived discomfort (10.1007/s00134-010-1902-9, 10.1007/s00134-017-4991-x, 10.1007/s00134-018-05511-y).

The IPREA study group also proposed adding two items to the initial version of the IPREA questionnaire, leading to an 18-item version (ICU-related feelings of depression and ICU-related breathing discomfort). The 18-item version of IPREA was validated in a multicenter, cluster-randomized, controlled, two-parallel group French study. A total of 994 patients were included. This study confirms that this French questionnaire asking about patients’ self-perceived ICU discomforts are reliable and valid (10.1186/s12955-019-1101-5).

Text added in Materials and methods :

The 18-item French-language IPREA questionnaire (Supplementary Table 1), which was validated by the IPREA group in a large sample of critically ill patients hospitalised in 34 French ICUs (12).

Clinical Impression at ICU Discharge:

The study mentions the clinical impression of ICU physicians and psychologists at discharge to predict psychiatric symptoms three months post-ICU stay. However, the methodology lacks sufficient detail regarding how these predictions were made. Specifically:

Were predictions based solely on clinical interviews or supported by structured data?

What criteria or scoring methods were used to assess the risk?

These details are essential for evaluating the reliability and reproducibility of the findings.

We agree with your suggestions

Text added in materials and methods:

ICU physicians and psychologists met the patient for a clinical interview to assess the occurrence of psychiatric symptoms or not (anxiety, depression or PTSD; associated or not) before scoring his or her self-perceived discomfort. They used clinical experience and patients’ clinical data only (health status at ICU admission, condition of admission, number of stressful procedures, ICU length of stay, memories during sedation and frightening experiences).

Reviewer #1: Statistical Analysis

T-test vs. Welch’s Test:

Welch's t-test is generally considered more robust when there are concerns about equal variances between groups. The manuscript does not address whether the assumption of equal variances was tested. It would be more appropriate to use either Welch's t-test exclusively or clearly justify the choice of the standard t-test.

We opted to use the Mann-Whitney U test instead of Welch's t-test for comparing our continuous variables because it provides an even more flexible approach, as it does not rely on variance assumptions at all. Additionally, for small to moderate sample sizes, non-parametric methods often yield greater statistical power.

Categorical Variables:

For categorical variables, Fisher's exact test is preferable due to its robustness, especially for smaller sample sizes. Using only Fisher’s exact test could simplify and strengthen the statistical approach.

We agree with your suggestion.

We have adjusted the p-values by using Fisher’s Exact Test instead of the Chi-square test as recommended.

Text added in result : p value in Table 3

Multivariate Model Construction:

It is unclear whether the multivariate models were built using a forced-entry method or another approach (e.g., stepwise selection). The manuscript should explicitly state the method used for model construction.

The selection of variables included in our multivariate model was based on both statistical recommendations and clinical and scientific considerations. Considering the limited sample sizes, we followed the rule of thumb of having approximately 10 outcome events per predictor to avoid overly complex models and overfitting (Peduzzi et al.,1996). Given that our sample included n = 44 events, we adhered to this principle and limited the model to four variables to ensure reliable estimates.

Also, the choice of the four variables was guided by the univariate analysis p<0.2 and clinical relevance on known confounding factors. To assess the stability and robustness of our findings, we also tested multiple models, varying the included covariates. This sensitivity analysis helped to confirm that our main conclusions remained consistent across different model specifications.

Text added in materials and methods:

Multivariate logistic regression model was constructed following the commonly recommended guideline of maintaining approximately 10 outcome events per predictor variable to mitigate the risk of model overfitting and excessive complexity (15). The selection of variables was based on a univariate analysis threshold (p < 0.2) combined with clinical relevance and prior evidence on known confounders.

Reviewer #1: Results

Cutoff Values:

The rationale for using cutoff values instead of continuous variables is not provided. Continuous data might offer more granularity and statistical power. Additionally, the precision of the cutoff values seems questionable and should be justified statistically.

The categorization of variables was chosen to enhance clinical interpretability and ensure homogeneous risk groups. Keeping variables continuous in multivariate models did not reveal a clear effect, making categorization a more suitable approach.

Additionally, models with continuous variables showed poorer fit (higher AIC and BIC), further justifying categorization. While we acknowledge the potential gain in statistical power with continuous variables, this was not a limitation here, as a significant effect was detected using categorized variables.

For reference, the table below presents the models with variables in their continuous form.

Table 2. Effect of variables on the PICS occurence at 3 months in multivariate analysis

OR (95% CI) p value

Model 1 (AIC=139.8, BIC=150.5)

IPREA score 0.98 (0.95 – 1.01) 0.248

Psychiatric history 3.4 (0.94 – 12.39) 0.06

Age 1.03 (1 – 1.06) 0.06

Model 2 (AIC=141.5, BIC=154.9)

IPREA score 0.98 (0.95 – 1.01) 0.29

Psychiatric history 3.6 (1 – 13.27) 0.054

Age 1.04 (1 – 1.07) 0.024

Health status with limitation of activity a 2 (0.7 - 5.67) 0.20

Model 3 (AIC=143.1, BIC=159.2)

IPREA score 0.99 (0.95 – 1.02) 0.43

Psychiatric history 3.48 (0.94 – 12.85) 0.06

Age 1.04 (1 – 1.1) 0.02

Health status with limitation of activity a 2.2 (0.75 – 6.4) 0.15

Female sex 1.85 (0.74 - 4.6) 0.19

Text added in material and methods:

Variables were categorized to improve clinical interpretability and define homogeneous risk groups, with cutoff values determined using receiver operating characteristic (ROC) curves and the Youden index.

Reason for Three Models:

The reason for constructing three separate models is not explained. Furthermore, no model fit indices (e.g., AIC, BIC, or R-squared) are reported, making it impossible to determine which model performs best.

To assess the stability and robustness of our findings, we tested multiple models, varying the included covariates. This sensitivity analysis helped to confirm that our main conclusions remained consistent across different model specifications. We have provided AIC and BIC for each multivariate model to facilitate model comparison and assess which model performs best.

Text added in methods:

To evaluate the stability and robustness of our findings, we conducted sensitivity analyses by testing multiple model specifications with varying covariate selections. This approach allowed us to confirm the consistency and reliability of our main conclusions across different analytical frameworks. AIC and BIC values were also reported for each multivariate model to assess model fit and complexity.

Text added in result:

AIC and BIC in Table 2

Expression of Primary Outcome:

The phrase "achieved primary outcome" implies a positive connotation, which might not be appropriate in this context. A more neutral expression, such as "primary outcome was assessed" or "primary outcome was observed," would be preferable.

We agree about your suggestion. We applied the correction in your manuscript.

Text added in result and table:

The primary outcome was observed in 44 (40%) patients of our cohort study.

The patients for whom the primary outcome was observed had their quality of life reduced according to the WHOQOL-BREF questionnaire.

Statistical Method for Clinical Impression:

The statistical method used to analyze "the clinical impression of the physician and psychologist" is not mentioned in the Methods section. This lack of clarity prevents a proper evaluation of the robustness of the results.

We explicit the method in the article and reword the sentence.

Text added in methods:

Agreement between the patient's feeling of distress as assessed by the physician or psychologist at ICU discharge to the occurrence of psychiatric symptoms at 3-months was evaluated using the Cohen's kappa coefficient, which measures inter-rater reliability beyond chance, with p-values assessing its significance against zero.

Response to Reviewer #2

Dear Reviewer,

The new version of our manuscript includes all your suggestions.

We hope that these responses will satisfy you.

Sincerely yours

Reviewer #2: General Comment

The manuscript provides valuable insights into the role of self-perceived discomfort during ICU stays as a predictive factor for post-ICU psychiatric disorders. The methodology is sound, and the statistical analyses are appropriate for the study objectives. However, some areas require further clarification and refinement to enhance the robustness and interpretability of the findings.

We hope to carefully clarify your suggestions.

Reviewer #2: Specific Comments

Sample Size and Attrition

The final analysis included 109 participants, which is below the minimum required sample size of 153 calculated during the study design phase. While the study provides meaningful results, the reduced sample size may limit the statistical power and generalizability of the findings.

Furthermore, the high attrition rate (37%, with 64 participants excluded due to loss to follow-up or incomplete data) is a concern. Although the authors report no significant differences between included and excluded participants, the impact of this attrition on the representativeness of the cohort should be discussed in more detail. It is recommended that future studies adopt measures to minimize attrition, such as more robust follow-up mechanisms or alternative methods for data collection.

Thank you for your comment. We agree with you that high attrition is a comment. In your study, we observed 26% (46) of excluded patients for missing data due to no respond about the 3-month follow up. It is more than expected in your methodology (20% initially planned). However, the proportion of included patients with complete follow-up data at 3-months was slightly over 50% and, although comparable to that of many studies on post-ICU psychiatric symptoms (4,5,16). Moreover, we observed 44 patients for whom the primary outcome was observed. That in line of our patient number needed to show an area under the curve (AUC) of 0.7 with a 95% CI of 0.19 (153 patients required: 46 PICS and 107 no PICS).

Text added in discussion:

We observed 46 (26%) excluded patients for missing data, due to no respond after 3-month of follow up (20% initially planned). However, the proportion of included patients with complete follow-up data at 3-months was slightly over 50%? This was comparable to many studies on post-ICU psychiatric symptoms (4,5,17). Moreover, we observed 44 (25%) patients for whom the primary outcome was reached. That was in line with our required number of patients to show an AUC of 0.7 with a 95% CI of 0.19 (153 patients required: 46 with PICS and 107 without PICS).

Adjustment for Confounders

The study effectively uses multivariable logistic regression to adjust for confounding factors, including age, pre-existing psychiatric disorders, and IPREA scores. However, quality of life (QoL) was evaluated as a secondary outcome and not included as a potential confounding factor in the analysis. Considering that QoL is closely associated with both ICU experiences and psychiatric outcomes, its exclusion as a covariate may limit the understanding of its role in mediating or moderating the relationship between self-perceived discomfort and psychiatric outcomes. Future studies could benefit from incorporating QoL as a covariate to disentangle its effects and provide more nuanced insights.

We are in line about your thinking. The increasing number of ICU survivors means that future studies and a continuous quality health care improvement strategy must focus on improving long-term patients' outcomes and quality of life after ICU stay. However, in our study, we assessed the quality of ICU patients’ life after the ICU stay in order to highlight the impact of self-perceived discomfort and also post intensive care syndrome on quality of life. This data does not include in our multivariate analysis because she was observed not at ICU discharge but 3-month after. Measuring quality of life at ICU discharge seems not suitable for us.

If we incorporated quality of life as an adjustment variable in the multivariate models, the conclusions remain unchanged from those presented in our study. Since AIC and BIC values were better without this variable, we propose maintaining the original results.

Table 2. Effect of variables on the PICS occurence at 3 months in multivariate analysis

OR (95% CI) p value

Model 4 (AIC=49.5, BIC=62.9)

IPREA score ⩾ 13 3.7 (1.4 - 10) 0.009

Psychiatric history 3.5 (0.9 - 13.4) 0.06

Age ⩽ 52 y 2.6 (1 - 6.8) 0.06

QoL 0.7 (0.4 – 1.3) 0.32

Model 5 (AIC=66, BIC=82.1)

IPREA score ⩾ 13 3.8 (1.4 - 10) 0.01

Psychiatric history 3.6 (0.9 - 14) 0.06

Age ⩽ 52 y 3.2 (1.1 - 9.5) 0.03

Health status with limitation of activity a 1.8 (0.6 - 5.2) 0.3

QoL 0.8 (0.4 – 1.4) 0.38

Model 6 (AIC=87.3, BIC=106.1)

IPREA score ⩾ 13 3.4 (1.2 - 9.1) 0.02

Psychiatric history 3.5 (0.9 - 13.3) 0.07

Age ⩽ 52 y 3.6 (1.2 - 11.1) 0.02

Health status with limitation of activity a 2 (0.7 – 5.7) 0.22

Female sex 1.7 (0.7 - 4.5) 0.3

QoL 0.8 (0.4 – 1.4) 0.35

On the other hand, physical health and function, such as ICU-acquired weakness at ICU discharge, may also affect mental health outcomes. This data was not reported in our study. We included this comment in our discussion.

Text added in discussion:

Physical health and function, such as ICU-acquired weakness, may also affect mental health outcomes and may limit the understanding of the relationship between self-perceived discomfort and psychiatric outcomes (25).

Role of QoL in the Analysis

While the study highlights the impact of psychiatric symptoms on QoL, it does not assess whether baseline QoL or cha

---

## [Editor Report · Decision Letter 1]

22 Apr 2025

Impact of self-perceived discomfort in critically ill patients on the occurrence of psychiatric symptoms in post-intensive care syndrome (PICS): A prospective observational study

PONE-D-24-52846R1

Dear Dr. Ronflé,

We’re pleased to inform you that your manuscript has been judged scientifically suitable for publication and will be formally accepted for publication once it meets all outstanding technical requirements.

Kind regards,

Michihiro Tsubaki

Academic Editor

PLOS ONE

Additional Editor Comments (optional):

I have confirmed sufficient corrections to the comments.

I am pleased that this paper will be published.
---

## [Editor Report · Acceptance letter]

PONE-D-24-52846R1

PLOS ONE

Dear Dr. Ronflé,

I'm pleased to inform you that your manuscript has been deemed suitable for publication in PLOS ONE. Congratulations! Your manuscript is now being handed over to our production team.

Kind regards,

on behalf of

Dr. Michihiro Tsubaki

Academic Editor

PLOS ONE